# Sentinel Lymph Node Staging with Indocyanine Green for Patients with Cervical Cancer: The Safety and Feasibility of Open Approach Using SPY-PHI Technique

**DOI:** 10.3390/jcm10214849

**Published:** 2021-10-21

**Authors:** Mustafa Zelal Muallem, Ahmad Sayasneh, Robert Armbrust, Jalid Sehouli, Andrea Miranda

**Affiliations:** 1Department of Gynecology with Center for Oncological Surgery, Charité—Universitätsmedizin Berlin, Corporate Member of Freie Universität Berlin, Humboldt-Universität zu Berlin, and Berlin Institute of Health, Berlin, Virchow Campus Clinic, Charité Medical University, 13353 Berlin, Germany; robert.armbrust@charite.de (R.A.); jalid.sehouli@charite.de (J.S.); andrea.miranda@charite.de (A.M.); 2Department of Gynecological Oncology, Surgical Oncology Directorate, Guy’s and St Thomas’ NHS Foundation Trust, School of Life Course Sciences, Faculty of Life Sciences and Medicine, King’s College London, Westminster Bridge Road, London SE1 7EH, UK; Ahmad.sayasneh@gstt.nhs.uk

**Keywords:** sentinel lymph node, cervical cancer, radical hysterectomy, indocyanine green, SPY-Portable Handheld Imager, SPY-PHI

## Abstract

**Simple Summary:**

While several studies have been conducted on the safety and efficacy of sentinel lymph node technique during minimally invasive radical hysterectomy and indicated that using indocyanine green alone is a better tracer agent, there is now high unmet medical need and growing demand for more data about sentinel lymph node detection and the most suitable tracer in open surgery for cervical cancer, especially after the publishing of the of Laparoscopic Approach to Cervical Cancer (LACC) Trial data. The aim of this study is to assess the feasibility and safety of sentinel lymph nodes with indocyanine green in cervical cancer patients undergoing radical hysterectomy in open surgery and to compare the detection rates of this tracer in the open versus laparoscopic approaches.

**Abstract:**

(1) Background: Sentinel lymph node staging (SLN) with indocyanine green (ICG) in cervical cancer is the standard of care in most national and international guidelines. However, the vast majority of relevant studies about the safety and feasibility of this method are conducted on minimally invasive surgery; (2) Methods: This study is a retrospective analysis of a retrospective collected database of 76 consecutive patients with cervical cancers, who were operated laparoscopically (50%), or laparotomy (50%). Sentinel nodes were defined as the ICG-positive pelvic nodes in the first and second echelons. False negative cases were defined as positive non-sentinel lymph nodes despite successful sentinel mapping or failed mapping bilaterally by per-patient assessment or unilaterally by pelvic sidewall assessment; (3) Results: Regardless of the approach (open or laparoscopic), the SLN technique achieved a total sensitivity, specificity, and negative predictive value (NPV) of 94.7%, 98.6%, and 94.7%, respectively in the entire sample. The bilateral detection rate was as high as 93.4% with identical results in both approaches. The sensitivity and NPV for SNL in open surgery was found to be similar to minimal access surgery; (4) Conclusions: ICG and SPY-PHI technique is a reliable tool to detect sentinel lymph nodes in cervical cancer during laparotomy.

## 1. Introduction

Lymph node metastatic spread is the most important prognostic factor in early-stage cervical cancer. The reported survival rates for patients with FIGO-stage I cervical cancer are between 80% and 98%. However, the five-year survival of these patients can drop significantly to 50% when the lymph nodes are involved [1]. This led the International Federation of Gynaecology and Obstetrics (FIGO) to considering all nodal positive cases as stage IIIC in its updated version as lymph node involvement is associated with a worse prognosis [2].

The preoperative assessment of the lymph nodes involvement remains less accurate than pathologic evaluation of lymph nodes even when using PET-CT, which is the preferred imaging modality to assess for metastatic disease [3,4,5]. To detect nodal metastasis greater than 10 mm, PET-CT has shown a better detection rate compared to CT and MRI, with a false negativity of only 4–15% [6,7,8]. By contrast, sentinel lymph node staging (SLN) has proven accurate in identifying lymph node metastasis with a detection rate of 95% and sensitivity of 100% in up to stage IB1 cervical cancer [9] and is now increasingly adopted. However, as shown in a recent extensive international survey on the current practice patterns of SLN in cervical cancer, SLN is not routinely used in the management of cervical cancer [10].

Indocyanine green (ICG) has been shown to be superior to blue dye or radioactive tracer in SLN bilateral detection rate in minimal access surgery [11,12]. However, its use in open surgery is limited to few case reports or case series [13,14,15,16,17,18,19]. Following the Laparoscopic Approach to Cervical Cancer (LACC) Trial published data [20], international societies, albeit with some bias, have adopted open surgery as the recommended approach [21,22]. Therefore, there is now a higher demand to offer more reliable data on the feasibility and safety of SLN in open surgery for cervical cancer, especially as using ICG has proven to be the best tracer in this disease. Different studies assessed the role of SLN in open radical surgery with blue dye or radioactive tracer [23,24,25,26,27], but only a few with ICG [13].

This study aims primary to assess the feasibility of SLN with ICG in cervical cancer patients undergoing open radical hysterectomy with SLN and compare its detection rate to the laparoscopic approach. The secondary objective of this study is to assess the safety of ICG as a tracer to detect SLN in open operated cervical cancer patients.

## 2. Materials and Methods

This study is a retrospective analysis of a prospective collected database. After institutional board review approval (under registration number EA1/174/14), consecutive patients who underwent radical hysterectomy at our institution between January 2014 and June 2021 were included in this analysis. All data were documented in a validated data bank. Written informed consent was obtained before collecting clinical data. Cancer staging in the study was adjusted to The International Federation of Gynaecology and Obstetrics (FIGO) classification from 2019 [2]. Patients were thoroughly counselled about the different possible options of treatment. In particular, women with greater than or equal to stage IB3 disease were informed of the concomitant chemo-radiation to be the standard treatment, and that radical hysterectomy represented an experimental option in their case.

Patients were eligible if they had squamous cell carcinoma, adenocarcinoma, or adenosquamous carcinoma of the cervix. Patients with clinically suspected distant metastasis, previous treatments, or allergy to iodine were excluded. Once written informed consent was confirmed, patients underwent laparoscopic or abdominal radical surgery according to the surgeon indication and patients’ preferability. The laparoscopic approach was the standard of care in our institution until the publication of the LACC Trial in 2018 [20]. After that point in time, we have changed our standard approach favouring the open surgery [28]. All procedures were performed by the first author (M.Z.M). A radical hysterectomy was performed in patients who did not wish to retain fertility, while a radical trachelectomy was offered to patients who desired to preserve fertility with a tumor diameter ≤2 cm. Preoperative histology and imaging (magnetic resonance imaging (MRI) scan, chest X-ray, whole-body positron emission tomography (PET)/computed tomography (CT) scan) details were collected.

At the time of surgery, after abdominal entry, all patients underwent intraoperative SLN with a standardized cervical injection of 1 cc of ICG 1.25 mg/mL in equally divided aliquots, both superficial (submucosal) and deep (1 cm into the stroma), at 3 and 9 o’clock. About 20–30 min after cervical injection, pelvic retroperitoneal space was opened. Lymph nodes were assessed with a near infra-red (NIR) camera of the PINPOINT system (Stryker, Kalamazoo, Michigan, US) in the case of a laparoscopic approach (Figure 1 and Appendix A); SLN in open surgery was detected with the SPY-Portable Handheld Imager (SPY-PHI) (Stryker, Kalamazoo, Michigan, US) as illustrated in Figure 2 and showed in Appendix A. Both cameras are designed to provide surgeons with real-time visualization of tissue perfusion and lymphatic vessels and nodes.

Sentinel nodes were defined as the ICG-positive pelvic nodes in the first and second echelons; retroperitoneal spaces were explored, and sentinel nodes were removed, labelled (according to the side and anatomical site: external iliac, internal iliac, obturator, common iliac, and parametrial), and sent to pathology for frozen section. The SLN was followed in most cases by pelvic lymphadenectomy with or without para-aortic lymphadenectomy as SLN still not the standard approach according to the German guidelines [29]. All mapped sentinel nodes were analysed with the ultra-staging concept. To statistically compare open and laparoscopic SLN performance, patients were divided according to surgical approach. False negative cases were defined as positive non-sentinel lymph nodes despite successful sentinel mapping or failed mapping bilaterally by per-patient assessment or unilaterally by pelvic sidewall assessment. The radical hysterectomies were performed nerve-sparingly according to our technique described in a previous study [28]. The radicality was adjusted to the tumor volume, localization, infiltration in the vagina, and FIGO-stage in accordance with the Muallem classification of radical hysterectomy [30]. The statistical analysis was performed at the Charité Medical University Berlin. Frequency counts and percentages were used to describe categorical variables, and continuous variables were summarized the median and range. All *p*-values reported were two-sided, and a *p*-value < 0.05 was considered statistically significant. Analysis was computed using SPSS version 26.0 (IBM Corporation 2018, Armonk, NY, USA: IBM Corp.).

## 3. Results

Seventy-six patients who met the inclusion criteria were included in the study; three were excluded: two had neuroendocrine cervical tumours, and one was previously treated with radiotherapy. No patient had any allergy to iodine. Thirty-eight patients (50%) underwent open surgery and the other thirty-eight patients (50%) underwent laparoscopic sentinel lymph nodes staging. Clinical and pathological characteristics of the included patients are summarized in Table 1. Seventy patients (92%) underwent radical hysterectomy type C1 (Querleu-Morrow classification [31]), while a radical trachelectomy was performed in 6 (8%) patients who wished fertility to be preserved and were eligible to have the procedure.

The majority of patients had the final histology as squamous cell carcinoma (82.9%). In 47.4% of cases, the tumor diameter was more than 40 mm, and in 32.9%, it was between 20 mm and 40 mm. Patients who underwent an open approach reported a higher FIGO stage with 21 patients (55.3%) having a FIGO stage ≥ IB3 compared to 15 (39.5%) in the laparoscopic group. Furthermore, the laparotomic group showed a higher rate of grade 3 (44.7% vs. 31.6% for laparoscopic approach) and positive LVSI (57.9% in open surgery vs. 36.8% for laparoscopy). A larger median tumor diameter was found after open surgery compared to laparoscopy (41.6 mm vs. 38 mm, respectively; *p* = 0.003). Table 1 illustrates the patients and tumor characteristics.

Seventy-one (93.4%) patients received a successful bilateral sentinel lymph node staging, whereas the mapping did not show any sentinel lymph node and therefore was considered unsuccessful in three patients (4%). In two patients (2.6%), the ICG could reveal sentinel lymph nodes only unilaterally. Hence, the detection rate in this study was as high as 94.7%, with comparable results for the open approach (96%) and for minimal access surgery (93.4%). The bilateral detection rate was also as high as 93.4% with identical results in both approaches (94.7% for open surgery and 92.1% for minimal invasive surgery, *p* = 1.000). Overall, 782 sentinel lymph nodes were revealed, retrieved, labelled, and sent to histopathology for frozen section. The median number of resected sentinel lymph nodes per patient was 10.7 lymph nodes. However, when patients with detected sentinel lymph nodes only are considered, the median number of resected sentinels was 5.4 sentinel lymph nodes per pelvic sidewall. No difference in the median number of retrieved sentinel lymph nodes was found between the two approaches, with 11 lymph nodes resected per patient on average for both groups, ranging from 2 to 29 SLN for laparotomy and 1 to 33 SLN for laparoscopy).

The median number of retrieved lymph nodes was 52 nodes per patient (8–124) in the entire cohort, with 56 (12–124) nodes after open surgery and 47 (8–100) after minimal access procedures. No allergic reactions to ICG or any other side-effect or ICG-related complication were reported.

More than two-thirds of these patients (71% of the whole sample, 76% of patients with successful bilateral SLN) had no metastases in lymph nodes. In 17 cases (22.4% of the whole sample, 23.9% of patients with successful bilateral SLN) metastases were found at frozen section analysis, while in two cases (2.8%) the lymph nodes involvement was detected first during histopathological ultra-staging. In one patient only, who underwent a laparotomy, we intraoperatively retrieved an enlarged non-sentinel lymph node in paracolpium which was positive and considered as a positive non-sentinel lymph node (false negative).

The SLN technique achieved a total sensitivity, specificity, and negative predictive value of 94.7%, 98.6%, and 94.7%, respectively in the entire sample. When calculated per pelvic sidewall, the sensitivity of SLN was 94%, specificity was 99.3%, and the negative predictive value was 94%. Hence our practice was to perform a pelvic lymphadenectomy regardless of the results of SLN; failed mapping cases may not be counted as false negativity, which will increase the sensitivity of the procedure with ICG per pelvic sidewall to 99.3%, specificity to 99.3%, and the negative predictive value to 99.3%. When using SPY-PHI-technique and ICG in open surgery and when failed mapping is regarded as false negativity, the sensitivity specificity and negative predictive value of SLN were 94.7%, 97.4%, and 94.9% respectively.

The most common site of SLN was the common iliac region at the right and left pelvic sidewalls (77.5% and 67.6%, respectively) in the laparoscopic and laparotomic ICG-mapping (77.5% and 67.6%, respectively). This is followed by the external iliac regions (29.6% on the right side and 32.4% on the left side), then the obturator fossa (21% on the right side and 31% on the left side). Table 2 presents the distributions of sentinel lymph nodes in both groups.

## 4. Discussion

There is accumulating evidence in the literature to support the safety and feasibility of lymphatic mapping and sentinel lymph nodes staging in cervical cancer [12,13,16,32]. Notwithstanding the large tumor size up to 4 cm, there is still a good detection rate of SLN reported in a few studies; The SENTICOL I (Ganglion Sentinelle dans le Cancer du Col) study showed a 92% sensitivity and 98.2% negative predictive value for node metastasis detection in patients with cervical cancer who underwent technetium-99 lymphoscintigraphy and Patent Blue injection followed by the laparoscopic lymph node mapping [33]. No false-negativity was reported in the patients with bilaterally detected sentinel lymph nodes (76.5%). In comparison with technetium and the blue dye, this detection rate could be further improved by using the ICG technique. Retrospectively reviewed data from five European centres have already shown ICG to be superior to 99 m technetium and blue dye with a detection rate reaching 100% and with better visualization of the bilateral lymph drainage pathways (99%) [34,35]. A new Italian study demonstrated that there is no significant difference in ICG SLN diagnostic performance in cervical cancer between open and minimal access surgery, with a sensitivity and negative predictive values of 83.3% and 95.0%, respectively in open access surgery, and 92.9% and 97.5%, respectively in minimal access surgery [13]. These critical results emphasize the high unmet medical need and growing demand for more data on SLN detection using ICG in open surgery for cervical cancer, in particular following the LACC-trial findings [20], which have shifted our practice towards open access with a few exemptions. Our findings support the previously mentioned results and display a very high detection rate (overall 96%, and bilateral detection rate of 94.7%) in open surgery. To the best of our knowledge, this study included the largest collection of patients undergoing laparotomy for cervical cancers measuring up to 4 cm in diameter with ICG and SPY-PHI technique to detect sentinel lymph nodes.

In our opinion, the very good detection rates, sensitivity, and negative predictive values in this study may be attributed to the first and second echelons resection of sentinel lymph nodes, which led to the high median number of retrieved sentinel lymph nodes per pelvic sidewall. On the other hand, the most frequent site of SLN mapping was external iliac, followed by obturator region in the most studies considering the first labelled SLN only [13,36,37]. In our study was the common iliac region the most common site for SLN lymph nodes even because we retrieved not only the first but too the second echelon of labelled lymph nodes.

In spite of the prospectively collected data, the main limitation of our study is the retrospective analysis of this data. To this extent, the study group is planning the start of a randomized controlled trial soon to evaluate prospectively the safety and feasibility of ICG and SPY-PHI in detecting sentinel lymph node intraoperatively for patients with up-to-four cm cervical cancer, and to compare the performance of ICG to the combination of technetium-99m and blue-dye (A Prospective randomized, open-label, monocentric trial to assess the safety and utility of ICG for lymph node mapping in open surgery using SPY Portable Handheld Imager (SPY-PHI) ^®^ in CERvical cancer patients. (SPYCER-trial)).

## 5. Conclusions

The sentinel lymph node mapping in open surgery using ICG and SPY-PHI technique has statistically significant higher detection rates, and comparable sensitivity and negative predictive value compared to minimal access surgery. ICG and SPY-PHI technique may be the most reliable tool to detect sentinel lymph nodes in cervical cancer during laparotomy. The results of the SPYCER-trial will supply more authentic information.

## Figures and Tables

**Figure 1 jcm-10-04849-f001:**
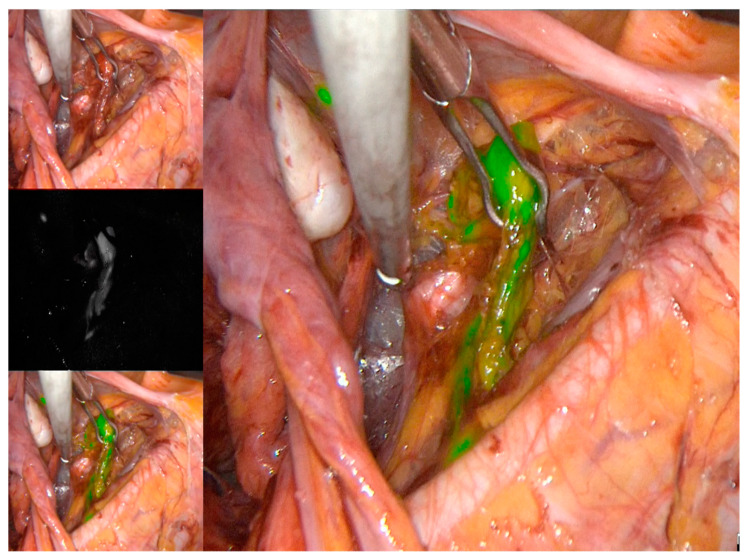
Anterior paracervical (sub-ureteral) pathway of lymphatic spread explored with ICG and near infra-red (NIR) camera of PINPOINT system in laparoscopic approach.

**Figure 2 jcm-10-04849-f002:**
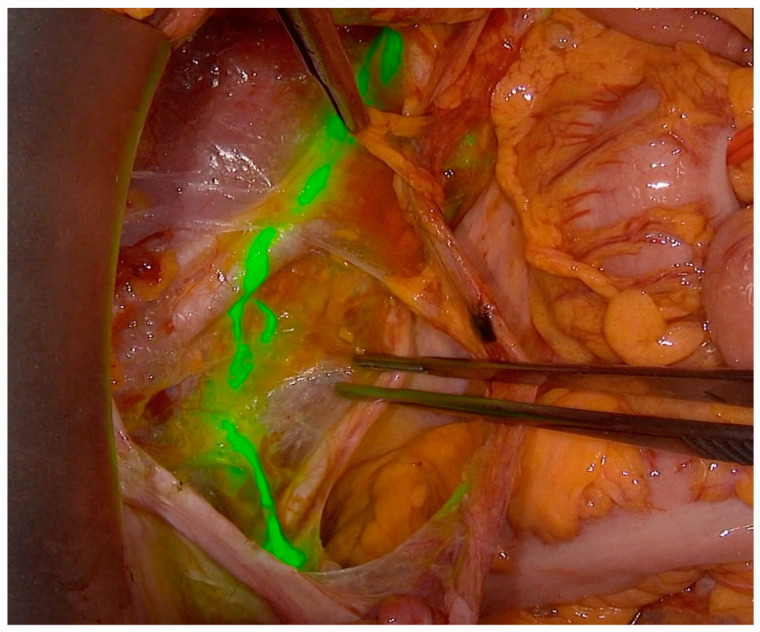
Posterior paracervical (supra-ureteral) pathway of lymphatic spread explored with ICG and SPY-Portable Handheld Imager (SPY-PHI) in open surgery.

**Table 1 jcm-10-04849-t001:** Patients and tumour characteristics.

Characteristic		N = 76, Median (Range, %)	Laparotomy = 38 (Range, %)	Laparoscopy = 38 (Range, %)	*p*-Value
Age (years)		45.6 (25.5–72.9)	46.1 (25.5–70.7)	45.6 (28.8–72.9)	0.192
BMI (kg/m^2^)		25.8 (17.6–43)	25.7 (18.2–43)	25.8 (17.6–34)	0.921
Preoperative conization		13 (17.1)	6 (16)	7 (18)	1.000
Type of surgery	Radical hysterectomy	70 (92)	38 (100)	32 (84.2)	0.025
	Radical trachelectomy	6 (8)	0	6 (15.8)	0.025
Sentinel lymph node	No	3 (4)	1 (2.6)	2 (5.2)	1.000
	Unilateral	2 (2.6)	1 (2.6)	1 (2.6)	
	Bilateral	71 (93.4)	36 (94.7)	35 (92.1)	
Number of SLN		11 (1–33)	11 (2–29)	11 (1–33)	0.926
Number of removed lymph nodes		52 (8–124)	56 (12–124)	47 (8–100)	0.766
Histology	Squamous cell cancer	63 (82.9)	31 (82)	33 (87)	0.754
	Adenocarcinoma	12 (15.8)	7 (18)	5 (13)	
	Adenosquamous	1 (1.3)	0	0	
Grading	1	2 (2.6)	0	2 (5.3)	0.277
	2	44 (58)	21 (55.3)	23 (60.5)	
	3	29 (38)	17 (44.7)	12 (31.6)	
	Unknown	1 (1.3)	0	1 (2.6)	
LVSI	Negative	38 (50)	16 (42.1)	22 (57.9)	0.073
	Positive	36 (47.4)	22 (57.9)	14 (36.8)	
	Unknown	2 (2.6)	0	2 (5.3)	
Tumor volume	≤20 mm	15 (19.7)	3 (8)	12 (32)	0.012
	>20–≤40 mm	25 (32.9)	17 (45)	8 (21)	
	>40 mm	36 (47.4)	18 (47)	18 (47)	
Mapping results	No SLN-involvement	54 (71)	27 (71)	27 (71)	1.000
	Positive in frozen section	17 (22.4)	9 (23.7)	8	
	Positive in ultra-staging	2 (2.6)	1 (2.6)	1 (2.6)	
	Non-sentinel *	1 (1.3)	1 (2.6)	0	
	Mapping without SLN	3 (3.9)	1 (2.6)	2 (5.2)	
Pathologic parametrial infiltration	Bilateral	18 (23.7)	12 (31.6)	6 (15.8)	<0.001
	Only right	7 (9.2)	7 (18.4)	0	
	Only left	7 (9.2)	5 (13.2)	2 (5.3)	
	Total	32 (42.1)	24 (63.2)	8 (21)	
FIGO	IA1	0	-	-	n.a.
	IA2	2 (2.6)	1 (2.6)	1 (2.6)	
	IB1	14 (18.4)	4 (10.5)	10 (26.3)	
	IB2	24 (31.6)	12 (31.6)	12 (31.6)	
	IB3	14 (18.4)	6 (15.8)	8 (21)	
	IIA1	3 (3.9)	1 (2.6)	2 (5.3)	
	IIA2	1 (1.3)	1 (2.6)	0	
	IIB	9 (11.8)	5 (13.2)	4 (10.5)	
	IIIA	1 (1.3)	1 (2.6)	0	
	IIIB	0	-	-	
	IIIC1	5 (6.6)	4 (10.5)	1 (2.6)	
	IIIC2	0	-	-	
	IVA	3 (3.9)	3 (7.9)	0	
	IVB	0	-	-	
TNM	pT1b1 pN0	13 (17.1)	4 (10.5)	9 (23.7)	n.a.
	pT1b1 pN1	6 (7.9)	3 (7.9)	3 (7.9)	
	pT1b2 pN0	6 (7.9)	3 (7.9)	3 (7.9)	
	pT1b2 pN1	2 (2.6)	0	2 (5.3)	
	pT2a1 pN0	5 (6.6)	2 (5.3)	3 (7.9)	
	pT2a1 pN1	0	-	-	
	pT2a2 pN0	1 (1.3)	0	1 (2.6)	
	pT2a2 pN1	6 (7.9)	4 (10.5)	2 (5.3)	
	pT2b pN0	18 (23.7)	12 (31.6)	6 (15.8)	
	pT2b pN1	14 (18.4)	10 (26.3)	4 (10.5)	
No residual tumour after conization		5 (6.6)	0	5 (13.2)	0.054

* This patient included in the no SLN-involvement and considered as false negative, n.a.: not applicable.

**Table 2 jcm-10-04849-t002:** Distributions of sentinel lymph nodes in open surgery and minimal-invasive approach.

N = 71			Laparotomy = 36 (%)	Laparoscopy = 35 (%)	*p*-Value
Location of SLN in the right pelvic sidewall ^†^	Parametrial	4 (5.6)	4 (11)	0	0.115
	Obturator	15 (21)	9 (25)	6 (17.1)	0.565
	Internal iliac	5 (7)	2 (5.6)	3 (8.6)	1.000
	External iliac	21 (29.6)	10 (27.8)	11 (31.4)	1.000
	Common iliac	55 (77.5)	27 (75)	28 (80)	1.000
Location of SLN in the left pelvic sidewall ^†^	Parametrial	3 (4.2)	2 (5.6)	1 (2.9)	1.000
	Obturator	22 (31)	8 (22.2)	14 (40)	0.206
	Internal iliac	8 (11.3)	4 (11.1)	4 (11.4)	1.000
	External iliac	23 (32.4)	12 (33.3)	11 (31.4)	1.000
	Common iliac	48 (67.6)	28 (77.8)	20 (57.1)	0.095

^†^ The rates calculated from 71 cases with bilateral mapping. Repetition (SLN in more than a region in the same patient) is possible.

## Data Availability

The data presented in this study are available on request from the corresponding author. The data are not publicly available due to the privacy and ethical policy of our institution.

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
