# Peer review of "Sentinel Lymph Node Staging with Indocyanine Green for Patients with Cervical Cancer: The Safety and Feasibility of Open Approach Using SPY-PHI Technique"

_jcm, 2021, doi:10.3390/jcm10214849_

Round 1

Reviewer 1 Report

I carefully read and evaluated the paper: “Sentinel Lymph Node Staging with Indocyanine Green for Patients with Cervical Cancer: The Safety and Feasibility of Open Approach using SPY-PHI Technique”. This is a retrospective study with the aim to investigate the sensitivity and negative predictive value for SLN in open surgery and the anatomical distribution of lymph node in patients affected by cervical cancer. This is an interesting study. 

I have the following considerations:

  • The “abstract” section is quite concise. It should be revised. In particular, the part on methods should be revised, because it is very limited. The type of study should also be specified. Lines 29-30 this sentence should be revised.
  • The primary and secondary objectives should be more precisely specified in the text.
  • In which year was sentinel node research performed at your centre? Since 2014 and always with ICG?
  • The mean number of sentinel lymph nodes removed is 11 for both approaches, up to a maximum of 33 lymph nodes. In these cases, perhaps a lymphadenectomy is performed. Maybe this data should be explained in more detail and the predictive values analysed on the basis of these numbers.
  • Are there differences in terms of intra- and post-operative complications?

Author Response

Dear reviewer of Cancers,

Thank you very much for reviewing our manuscript and for your valuable comments.

  • The “abstract” section is quite concise. It should be revised. In particular, the part on methods should be revised, because it is very limited. The type of study should also be specified. Lines 29-30 this sentence should be revised.
  • Author’s reply: thank you very much for the very good point. We revised the abstract completely taking all your notices into account. Unfortunately, we still have to restrict the abstract on 200 words only, therefore we had to reduce some sentences in background and results.
    1) Background: Sentinel lymph node staging (SLN) with indocyanine green (ICG) in cervical cancer is the standard of care in most national and international guidelines. However, the vast majority of relevant studies about the safety and feasibility of this method are conducted on minimally invasive surgery; (2) Methods: This study is a retrospective analysis of a retrospective collected database of 76 consecutive patients with cervical cancers, who were operated laparoscopically (50%), or laparotomy (50%). Sentinel nodes were defined as the ICG-positive pelvic nodes in the first and second echelons. False negative cases were defined as positive non-sentinel lymph nodes despite successful sentinel mapping or failed mapping bilaterally by per-patient assessment or unilaterally by pelvic sidewall assessment; (3) Results: Regardless of the approach (open or laparoscopic), t he SLN technique achieved a total sensitivity, specificity, and negative predictive value (NPV) of 94.7%, 98.6%, and 94.7%, respectively in the entire sample. The bilateral detection rate was as high as 93.4% with identical results in both approaches. The sensitivity and NPV for SNL in open surgery was found to be similar to minimal access surgery; 4) Conclusions: ICG and SPY-PHI technique is a reliable tool to detect sentinel lymph nodes in cervical cancer during laparotomy.
  • The primary and secondary objectives should be more precisely specified in the text.
  • Author’s reply: thank you very much for this valuable suggestion. We added this sentence to the end of introduction to follow your advice:

This study aims primary to assess the feasibility of SLN with ICG in cervical cancer patients undergoing open radical hysterectomy with SLN and compare its detection rate to the laparoscopic approach. The secondary objective of this study is to assess the safety of ICG as a tracer to detect SLN in open operated cervical cancer patients.

  • In which year was sentinel node research performed at your centre? Since 2014 and always with ICG?
  • Author’s reply: our institution has a long-erm experience with SLN, began with Tc99m and Isosulfan blue. Since 2012, we started to adapt ICG as the main tracer to detect SLN with or without the combination with other tracers and up 2014, we used to inject ICG only to detect SLN in cervical cancer.
  • The mean number of sentinel lymph nodes removed is 11 for both approaches, up to a maximum of 33 lymph nodes. In these cases, perhaps a lymphadenectomy is performed. Maybe this data should be explained in more detail and the predictive values analysed on the basis of these numbers.
  • Author’s reply: thanks for the very good point. As we started early to use SLN with ICG (before publishing the results of FIRES and FILM-trials) and included patients with high-risk cervical cancer (47% of cases had Tumor > 4 cm), we retrieved SLNs from the first and second echelons, whereas the most study on SLN consider only the resection of Lymph nodes of the first echelon. This led to the high number of SLNs in this study. We referred already in line 109-110 to the definition of SLN in this study (Sentinel nodes were defined as the ICG-positive pelvic nodes in the first and second echelons). The explanation of high number of retrieved SLN and its effect on results is already discussed in lines 233-236 (In our opinion, the very good detection rates, sensitivity and negative predictive values in this study may be attributed to the first and second echelons resection of sentinel lymph nodes, which led to the high median number of retrieved sentinel lymph nodes per pelvic sidewall.)

  • Are there differences in terms of intra- and post-operative complications?
  • Author’s reply: there was no statistical significant difference between both approaches in term of complication (except for wound dehiscence). The data on intra- and postoperative complications were presented in our previous paper: Muallem MZ, Armbrust R, Neymeyer J, Miranda A, Muallem J. Nerve Sparing Radical Hysterectomy: Short-Term Oncologic, Surgical, and Functional Outcomes. Cancers. 2020; 12(2):483

Reviewer 2 Report

thank you to giving me the chance to review this article

despite good overall merit, I have some comments:

  • did you perform sample size or power analysis for feasibility outcome (detection rate)?
  • you have a perfect 1:1 ratio. Did you include data from all consecutive patients respecting study criteria?
  • did you perform only one ICG injection per case or did you have some cases with multiple injection? 

Author Response

Dear reviewer of Cancers,

Thank you very much for reviewing our manuscript and for your valuable comments.

  • did you perform sample size or power analysis for feasibility outcome (detection rate)? 
  • Author’s reply: Thank you very much for the important point. This study  is a retrospective analysis of a prospectively collected database, therefor there was no sample size or power analysis. To the best of our knowledge, this study included the largest collection of patients undergoing laparotomy for cervical cancers measuring up to 4 cm in diameter with ICG and SPY-PHI technique to detect sentinel lymph nodes. Despite that, we did not claim that our study offers the best evidence of the safety and feasibility of ICG in open surgery for cervical cancer. For this reason, we plan the prospective randomized study with 150 patients, which will be enrolled in SPYCER trial.
  • you have a perfect 1:1 ratio. Did you include data from all consecutive patients respecting study criteria?
  • Author’s reply: Thank you for this point. the number of patients in every group is accidentally the same, as we mentioned in the methods (Line 77): consecutive patients who underwent radical hysterectomy at our institution between January 2014 and June 2021 were included in this analysis.
  • did you perform only one ICG injection per case or did you have some cases with multiple injection? 
  • Author’s reply: Thank you for this question. We injected ICG once per case in the cervix in equally divided aliquots, both superficial (submucosal) and deep (1 cm into the stroma), at 3 and 9 o’clock 20-30 min before opening the retroperitoneal space.

Round 2

Reviewer 1 Report

I thank the authors for their comments and revisions. 

Reviewer 2 Report

thank you for your replies

This manuscript is a resubmission of an earlier submission. The following is a list of the peer review reports and author responses from that submission.

Round 1

Reviewer 1 Report

The article presents really interesting issue for feasibility of open approach for ICG technique in SLN detection in the course of cervical cancer.

The article is well written, although there are some fields which must be improved.

In the introduction paragraph, it is given that the aim of the study is to asses the feasibility and safety of SLN detection. (71-72) Except for a short mention about no allergic reactions after ICG administration, no other aspects of safety are given including comparison of intra and postoperative complications. If the authors would like to assess the safety of this procedure there should be more data with complications included. 

In methodology section it is not clearly written, that this is a retrospective study. Please include.

It is not given what kind of statistical tests have been used.

The number of retrieved SLNs and all other pelvic lymph nodes is very high.  It is seems necessary to include the method of pathological work-out in the methodology section. There is still a question which should be discussed how many SLNs are still SLN, as taking large numbers of nodes can seriously affect lymphatic drainage.

Author Response

Dear reviewer of Cancers,

Thank you very much for reviewing our manuscript and for your valuable comments.

  1. In the introduction paragraph, it is given that the aim of the study is to asses the feasibility and safety of SLN detection. (71-72) Except for a short mention about no allergic reactions after ICG administration, no other aspects of safety are given including comparison of intra and postoperative complications. If the authors would like to assess the safety of this procedure there should be more data with complications included. 

Author’s reply: There were no allergic reactions or adverse events attributable to ICG in this study. The adverse events related to surgery were mentioned in another publication (Muallem MZ, Armbrust R, Neymeyer J, Miranda A, Muallem J. Nerve Sparing Radical Hysterectomy: Short-Term Oncologic, Surgical, and Functional Outcomes. Cancers. 2020; 12(2):483.) and was not in the focus of this study. The most considerable prospective randomized study about this subject (FILM trial) did not register any complication using ICG as a tracer. There is no known adverse event for ICG other than the potential allergic reaction.

  1. In methodology section it is not clearly written, that this is a retrospective study. Please include.

Author’s reply: we have now added the sentence (This study is a retrospective analysis of a prospectively collected database.) to the methods.

  1. It is not given what kind of statistical tests have been used.

Author’s reply: we added this sentence (The statistical analysis was performed at the Charité Medical University Berlin. Frequency counts and percentages were used to describe categorical variables, and continuous variables were summarized as median and range. All p-values reported were two-sided, and a p-value < 0.05 was considered statistically significant. Analysis was computed using SPSS version 26.0 (IBM Corporation 2018, Armonk, NY: IBM Corp.)

  1. The number of retrieved SLNs and all other pelvic lymph nodes is very high.  It is seems necessary to include the method of pathological work-out in the methodology section.

Author’s reply: The high number of SLN lymph nodes can be justified our definition of the sentinel lymph nodes as the ICG-positive pelvic nodes in the first and second echelons. Unlike most studies which consider only the first echelon.

The high number of retrieved lymph nodes is attributed to the radical resection of lymph nodes and not to the pathological work-out methodology.

  1. There is still a question which should be discussed how many SLNs are still SLN, as taking large numbers of nodes can seriously affect lymphatic drainage.

Author’s reply: we totally agree but we decided to resect the first and second echelon of SLN as we included high-risk tumors (> 4 cm) which made 47.4% of all study cases. In the discussion we have mentioned (In our opinion, the very good detection rates, sensitivity and negative predictive values in this study may be attributed to the first and second echelons resection of sentinel lymph nodes, which led to the high median number of retrieved sentinel lymph nodes per pelvic sidewall.)

We are very thankful for the valuable review.

Reviewer 2 Report

Cancers (ISSN 2072-6694)

Manuscript ID

cancers-1315208

Title

Sentinel Lymph Node Staging with Indocyanine Green for Patients with Cervical Cancer: The Safety and Feasibility of Open Approach using SPY-PHI Technique.

Authors

Mustafa Zelal Muallem * , Ahmad Sayasneh , Robert Armbrust , Jalid Sehouli , Andrea Miranda

Dear Editor

I have with great interest read the paper as of below. Although the LACC study, as well as in particular the Melamed US register study may suffer from bias regarding timing of the study related, and that national register studies have failed to verify differences in survival between open and MIS / robotic RH’s the results have raised concerns related the MIS approach, in particular in larger cervical cancers. It is of interest that two new RCT’s comparing RRH and ORH’s are recruiting, the RACC study, since almost two years and the ROCC study more recently. An issue with these studies may be differences in the accuracy of the SLN concept between approaches, potentially in favor of the MIS SLN approach.

Therefore, studies comparing SLN between open and MIS surgery is principally of interest.

There has been a great interest in SLN detection in uterine (endometrial and cervical) cancers over the years resulting in many hundred publications. Unfortunately, the absolute majority of these studies, including many of studies referred to, are retrospective and suffer from a lack of / or insufficient SLN-definition and methodological issues.

Prospective studies with the use of ICG ( proven to be the superior tracer) in MIS report up to 98% bilateral mapping/detection rates ( comparable with the present study), and up to 100% sensitivity although due to the rarity of the disease based on fewer cases with metastatic pelvic nodes as in studies on SLN in endometrial cancer using the same tracer (ICG) and injection site( Cervical).

I have the following concerns with the present paper.

  1. As I interpret the study is retrospective, comparing early MIS SLN results with later open SLN results. Apart from the risk of bias of data quality retrospectively retrieved from surgeries over 7 years, likely with many different surgeons involved, the fact that early MIS SLN result are compared with later Open surgery SLN results is a major bias as likely the experience has evolved over the years in favor of the open Group.
  2. The number and range of SLNs is unusually high; median 11 per patient (range 1-33), despite that presacral SLNs are not included ( which is a methodologic issue) implying a generous, rather that strict, definition of what is a SLN. The authors are urged to explain further how SLNs were defined, why the number is so high and how that may affect results and interpretation.
  3. The number of included patients is far below what is needed to compare sensitivity between approaches (10 and nine women with nodal involvement respectively). A sample size analysis for achieving significance is recommended for the discussion.
  4. The authors state that SLNs most often are localized in the common iliac position: This is in sharp contrast to prospective studies om cervical and endometrial cancer using ICG where this position, by far, is the least frequent position both for SLNs and SLN metastases. Please explain and relate to anatomically based prospective studies.
  5. Why was not presacral SLN defined (Lower paracervical lymphatic pathway?)
  6. In an increasing number of countries, based on MRI/ Sedlis criteria women with larger (Stage >=1b3 ) or mainly intracervical tumors tend to more often to be offered primary RCT rather that surgery. In this study, a very high percentage of positive nodes and parametrial involvement is reported. Please discuss how this may affect interpretation, sample size analysis, of the results in comparison with a situation where 10-12% of women are node positive and few have parametrial involvement.
  7. With the use of ICG the whole lymphovascular parametrium is usually green. Explain how SLNs were distingushed in that tissue.

In general, the study has a merit as it demonstrates the feasibility of SLN in open surgery. In my view the study should be rewritten as a sole description of the open approach with clarifications on questions as of above.

Based on its retrospective design, associated major risk of bias as described, and insufficient sample size the conclusions related comparison between approaches is not supported.

I wish the best for the attempted prospective study.

Author Response

Dear reviewer of Cancers,

Thank you very much for reviewing our manuscript and for your valuable comments.

  1. As I interpret the study is retrospective, comparing early MIS SLN results with later open SLN results. Apart from the risk of bias of data quality retrospectively retrieved from surgeries over 7 years, likely with many different surgeons involved, the fact that early MIS SLN result are compared with later Open surgery SLN results is a major bias as likely the experience has evolved over the years in favor of the open Group. 

Author’s reply: This study is a retrospective analysis of a prospectively collected database. The laparoscopic and open surgical modalities were both available options for patients and were performed over all the study period according to the surgical indication, and patients preference up till 2018. After publishing the results of LACC-trial, both modalities were still available to patients with favoring the open approach. All procedures were performed as standardized by the first author (M.Z.M.), who already had a long experience with the sentinel mapping technique. Therefore, we do not think that there is a significant bias of experience.
But we thank you for this point very much. To this extent, We have now included this point in the methods: Once written informed consent was confirmed, patients underwent laparoscopic or abdominal radical surgery according to the surgeon’s indication and patient’s preference. The laparoscopic approach was the standard of care in our institution until the publication of the LACC Trial in 2018 [20]. After that point in time, we have changed our standard approach favoring open surgery [28].

2.- The number and range of SLNs is unusually high; median 11 per patient (range 1-33), despite that presacral SLNs are not included (which is a methodologic issue) implying a generous, rather that strict, definition of what is a SLN. The authors are urged to explain further how SLNs were defined, why the number is so high and how that may affect results and interpretation

Author’s reply: Thanks for this important point. We already referred in lines 105-106 to the definition of SLN in this study (Sentinel nodes were defined as the ICG-positive pelvic nodes in the first and second echelons). The explanation of the high number of retrieved SLN and its effect on results is already discussed in lines 228-231 (In our opinion, the very good detection rates, sensitivity and negative predictive values in this study may be attributed to the first and second echelons resection of sentinel lymph nodes, which led to the high median number of retrieved sentinel lymph nodes per pelvic sidewall.)

3.- The number of included patients is far below what is needed to compare sensitivity between approaches (10 and nine women with nodal involvement respectively). A sample size analysis for achieving significance is recommended for the discussion. 

Author’s reply: To the best of our knowledge, this study included the largest collection of patients undergoing laparotomy for cervical cancers measuring up to 4 cm in diameter with ICG and SPY-PHI technique to detect sentinel lymph nodes. Despite that, we did not claim that our study offers the best evidence of the safety and feasibility of ICG in open surgery for cervical cancer. For this reason, we plan the prospective randomized study with 150 patients, which will be enrolled in SPYCER trial. The current paper is about a descriptive study with a retrospective analysis of a prospective collected database.

4.- The authors state that SLNs most often are localized in the common iliac position: This is in sharp contrast to prospective studies om cervical and endometrial cancer using ICG where this position, by far, is the least frequent position both for SLNs and SLN metastases. Please explain and relate to anatomically based prospective studies. 

Author’s reply: This is because of the different definitions of lymph nodes regions and the SLN lymph nodes, as we define the regions of lymph nodes according to the main vessels using the LION-Trial description as a standard definition for this study (right and left common iliac region: defined by aortic bifurcation, psoas muscle, iliac artery bifurcation, and the sacral os). In this way, the common iliac region in our study included both the common iliac region and the presacral region described in other studies.
Our definition of SLN as the ICG-positive pelvic nodes in the first and second echelons allows us to resect more SLN lymph nodes outside the common SLN-positions (reflecting only the SLN of the first echelon).
Our extended definition of SLN lymph nodes in our study was necessary as 47% of our patients have a tumor of more than 4 cm (increased metastatic risk).

We added this sentence to the discussion: On the other hand, the most frequent site of SLN mapping was external iliac, followed by obturator region in the majority of studies applied first labeled SLN mapping only [13, 36, 37]. Our study found that the common iliac region was the most common site for SLN lymph nodes because we retrieved both the first and second echelon of labeled lymph nodes.

  1. Why was not presacral SLN defined (Lower paracervical lymphatic pathway?). 

Author’s reply: the presacral SLN lymph nodes are included in common iliac lymph nodes. The definition of presacral SLN does not distinguish between right and left side, therefore we abandoned it.

  1. In an increasing number of countries, based on MRI/ Sedlis criteria women with larger (Stage >=1b3) or mainly intracervical tumors tend to more often to be offered primary RCT rather that surgery. In this study, a very high percentage of positive nodes and parametrial involvement is reported. Please discuss how this may affect interpretation, sample size analysis, of the results in comparison with a situation where 10-12% of women are node positive and few have parametrial involvement. 

Author’s reply: In our study, 47% of patients included had  > 4 cm tumor, which justifies the increased number of parametrial invasions and affected lymph nodes. Despite that, we were able to show a very good detection rate for SLN mapping in our collection, which goes in tandem with the results of other studies (Dostálek L, Zikan M, Fischerova D, Kocian R, Germanova A, Frühauf F, Dusek L, Slama J, Dundr P, Nemejcova K, Cibula D. SLN biopsy in cervical cancer patients with tumors larger than 2cm and 4cm. Gynecol Oncol. 2018 Mar;148(3):456-460. doi: 10.1016/j.ygyno.2018.01.001. Epub 2018 Feb 1. PMID: 29366509.).

  1. With the use of ICG the whole lymphovascular parametrium is usually green. Explain how SLNs were distingushed in that tissue.

Author’s reply: We have observed a different experience in our practice.  Please see Figure 1, and the supplementary videos to see how only the afferent lymph nodes in parametrium will be colored green. The spillage of green happens only by cutting through the lymph vessels in parametrium during the retroperitoneal preparation of SLN, which can be avoided in our experience.

In general, the study has a merit as it demonstrates the feasibility of SLN in open surgery. In my view the study should be rewritten as a sole description of the open approach with clarifications on questions as of above.

Author’s reply: thank you for this suggestion. The study was designed to compare the two arms. Therefore, restricting the study to one arm may not necessary lead to better results, in our humble opinion. However, we are very thankful for the valuable review and in particular for this suggestion. To this extent, and as mentioned above, we plan to conduct the prospective randomized study with 150 patients, which will be enrolled in SPYCER trial as illustrated in the paper.

Reviewer 3 Report

This manuscript seems to be a retrospective evaluation of a large collected dataset of cervical cancer pts. If this is the case, it should be mentioned. Specifically, the assessment of sentinel nodes was not the prime planned outcome of this dataset. 

methodology: are there differences in the temporal procurement of lpsc vs open sentinel nodes? I suspect prob, given the LACC trial. If so, there could be some bias, in that the more recent pts (open) would have higher detection rates than the earlier pts (lpsc).

The stats don't make sense. a bilateral detection rate of 94.7 vs 92.1 for open and lpsc respectively in 76 pts does NOT have a p<0.001. it is closer to 0.6. Thus the 2 techniques are similar, and one is not superior to the other. 

The number and location of sentinel lymph nodes is not in keeping with the literature. Most data would suggest approx. 2 sentinel lymph nodes/pelvic side, not 5.4. Increasing the number of sentinel lymph nodes removed/side increases the potential morbidity such as lymphedema (and cost), and is counter intuitive the whole concept. Furthermore, most authors have found obturator or internal iliac as the most common location, not common iliacs. One has to wonder if second and third echelon nodes are being removed and counted as first echelon. 

Author Response

Dear reviewer of Cancers,

Thank you very much for reviewing our manuscript and for your valuable comments.

This manuscript seems to be a retrospective evaluation of a large collected dataset of cervical cancer pts. If this is the case, it should be mentioned. Specifically, the assessment of sentinel nodes was not the prime planned outcome of this dataset.

Author’s reply: thank you for the good point, we added the sentence (This study is a retrospective analysis of a prospectively collected database.) to methods. In the discussion we already mentioned: In spite of the prospectively collected data, the main limitation of our study is the ret-rospective analysis of this data.

The assessment of Sentinel lymph node using ICG was primarily an aim of collecting this database.

Methodology: are there differences in the temporal procurement of lpsc vs open sentinel nodes? I suspect prob, given the LACC trial. If so, there could be some bias, in that the more recent pts (open) would have higher detection rates than the earlier pts (lpsc).

Author’s reply: This study is a retrospective analysis of a prospectively collected database.

All procedures were performed as standardized by the first author (M.Z.M.), who had already a long experience with sentinel mapping technique. Therefore, we do not think that there is a significant selection bias caused by operator’s experience. To this extent, we have clarified this in methods by adding this sentence: Once written informed consent wasconfirmed, patients underwent laparoscopic or abdominal radical surgery according to the surgeon indication and patient’s preference. The laparoscopic approach was the standard of care in our institution until the publication of the LACC Trial in 2018 [20]. After that point in time, we have changed our standard approach favoring the open surgery [28]. All procedures were performed as standardized by the first author (M.Z.M.).

The stats don't make sense. a bilateral detection rate of 94.7 vs 92.1 for open and lpsc respectively in 76 pts does NOT have a p<0.001. it is closer to 0.6. Thus the 2 techniques are similar, and one is not superior to the other.

Author’s reply: thank you very much for this important correction. The p-value was calculated right in the table 1. We changed the sentence to be: Hence, the detection rate in this study was as high as 94.7%, with comparable results for the open approach (96%) and for minimal access surgery (93.4%). The bilateral detection rate was also as high as 93.4% with identical results in both approaches (94.7% for open surgery and 92.1% for minimal invasive surgery, p= 1.000).

The number and location of sentinel lymph nodes is not in keeping with the literature. Most data would suggest approx. 2 sentinel lymph nodes/pelvic side, not 5.4. Increasing the number of sentinel lymph nodes removed/side increases the potential morbidity such as lymphedema (and cost), and is counter intuitive the whole concept. Furthermore, most authors have found obturator or internal iliac as the most common location, not common iliacs. One has to wonder if second and third echelon nodes are being removed and counted as first echelon.

Author’s reply: the high number of SLN lymph node because we considered the sentinel nodes as the ICG-positive pelvic nodes in the first and second echelons (most study consider only the first echelon). We considered to resect the second echelon of labeled lymph node as we included 47% of patients as high-risk tumors (> 4 cm).

We added this sentence to discussion: On the other hand, the most frequent site of SLN mapping was external iliac, followed by obturator region in the majority of studies applied first labeled SLN mapping only [13, 36, 37]. Our study found that the common iliac region was the most common site for SLN lymph nodes because we retrieved both the first and second echelon of labeled lymph nodes.

We are very thankful for the valuable review.

Round 2

Reviewer 1 Report

Dear Authors

Thank you for including my suggestions into the article. 

However, there are few aspects, which were just slightly touched. 

  1. The adverse events related to surgery were mentioned in another publication and was not in the focus of this study.

This stays in contrary in your aim of the study mentioned as below:

This study aims to assess the feasibility and safety of SLN with ICG in cervical cancer 70 patients undergoing open radical hysterectomy, moreover the title of the manuscript suggests that safety aspects of open LSN technique can be found within the text.

2. Still, It is not given what kind of statistical tests have been used. The tests which compare laparoscopic vs laparotomy group should be described. 

3. The other publications concerning radical lymphadenectomy of the pelvis (regardless of the disease) describe the maximum number of lymph nodes 52. It seems that radicality was comparable in all described papers, and the number of lymph nodes was at least halved. Therefore, the radicality of the procedure cannot explain your result and the pathology methodology should be included.

4. The same aspect is with the SLNs number - if the maximum number of SLN is 32 it means that for each side you have 16 nodes. SLNs means one-two in each "echelon". The detection rate is comparable to other authors, who describes much lower number of retrieved nodes. The detailed methodology of SLNs processing can be helpful in understanding the differences

Reviewer 2 Report

I have read the replies on the first revision and the revised paper.

The basic research question is very relevant and important; can a SLN procedure be accurately performed with enough sensitivity by open surgery, important as most studies on SLN in cervical cancer are performed by MIS.  It is of importance to be aware of that as of yet there is no prospective, well designed, methodologically well described and adequately powered study on SLN in cervical cancer. Such study would, to achieve reasonably narrow CIs for sensitivity need to be based on around 50 node positive patients. Studies with those prerequisites are ongoing.

The more relevant and important the research question, the higher are the demands of the research performed where there are two paramount questions. If the answer of these two questions, in particular question 1, is NO, the study should not be published or the conclusion rewritten or amended.

  1. Are the conclusions supported by the results?
  2. Are the results achieved in a methodologically adequate way?

I commend the authors for the work but I’m afraid that the conclusion regarding sensitivity and equality between methods is not supported by the results, simply because the number of node positive patients (those bearing information) is too low. The study in underpowered to address this question and the study would give the wrong impression to clinicians and principally risk health for patients.

I’m afraid that the authors do not sufficiently answer the review questions.

The bias due to the obvious fact that most MIS were performed in the initial phase (where experience with the SLN procedure was less) is problematic. The authors fail to give exact data on the proportion of MIS and OS over time, also the number of involved surgeons and case load for each of the surgeons involved. It is a known fact that case load is of importance when performing SLN. The claim that the procedures are performed according to the description of the first author is not enough. Potential biases should be highlighted and discussed.

The high average number of SLNs is a bit problematic. Obviously, the higher the number of SLN, the more the procedure approaches a full LND and the less likely it is that any differences between approaches will be identified. Is the number of SLNs as perceived by the surgeon (and defined by an intraoperatively used node protocol/anatomical chart, if so add this online) or as defined by the pathologist?

In all, the study in its present format, as a comparison between SLN between MIS and OP is underpowered and has methodological bias. The conclusion regarding first and foremost level of sensitivity and equality between MIS and OS is not supported sufficiently by data and my recommendation must therefore be REJECT!

I still do think that the study is important, and would reach better acceptance and impact, if it is rewritten and presented as a sole methodological study on SLN with open surgery without comparison with MIS and related conclusions. In that case, more detailed information on the methodology and protocols used should be added.

Reviewer 3 Report

changes to the manuscript are acceptable.